# Tuning the Photo-Luminescence Properties of WO_3_ Layers by the Adjustment of Layer Formation Conditions

**DOI:** 10.3390/ma13122814

**Published:** 2020-06-23

**Authors:** Milda Petruleviciene, Jurga Juodkazyte, Maliha Parvin, Alla Tereshchenko, Simonas Ramanavicius, Renata Karpicz, Urte Samukaite-Bubniene, Arunas Ramanavicius

**Affiliations:** 1Center for Physical Sciences and Technology, Sauletekio av. 3, LT-10257 Vilnius, Lithuania; milda.petruleviciene@ftmc.lt (M.P.); jurga.juodkazyte@ftmc.lt (J.J.); maliha.parvin@ftmc.lt (M.P.); simonas.ramanavicius@ftmc.lt (S.R.); renata.karpicz@ftmc.lt (R.K.); 2Department of Physical Chemistry, Institute of Chemistry, Faculty of Chemistry and Geosciences, Vilnius University, Naugarduko 24, LT-03225 Vilnius, Lithuania; alla_teresc@onu.edu.ua (A.T.); urte.samukaite-bubniene@chf.vu.lt (U.S.-B.); 3Department of Experimental Physics, Faculty of Mathematics, Physics and Information Technologies, Odesa National I.I. Mechnikov University, Pastera 42, 65023 Odesa, Ukraine

**Keywords:** tungsten (VI) oxide (WO_3_), sol-gel technique, cyclic voltammetry, sensors, photoelectrochemistry, time-resolved photoluminescence

## Abstract

In this research we have applied sol-gel synthesis for the deposition of tungsten (VI) oxide (WO_3_) layers using two different reductants (ethanol and propanol) and applying different dipping times. WO_3_ samples were characterized by X-ray diffraction (XRD), scanning electron microscopy (SEM), Fourier Transform Infrared spectroscopy (FTIR), photoluminescence (PL) and time-resolved photoluminescence decay methods. Photoelectrochemical (PEC) behaviour of synthesized coatings was investigated using cyclic voltammetry in the dark and under illumination. Formation of different structures in differently prepared samples was revealed and significant differences in the PL spectra and PEC performance of the samples were observed. The results showed that reductant used in the synthesis and dipping time strongly influenced photo-electrochemical properties of the coatings. Correlation between the morphology, PL and PEC behaviour has been explained.

## 1. Introduction

The demand for sensors with advanced analytical characteristics is constantly increasing [1]. Various metal oxides (TiO_2_, ZnO, SnO_2_, WO_3_, etc.) are widely used for numerous technological purposes including application in analytical signal transduction systems [2,3]. Nanostructured metal oxides are especially attractive in sensors due to their electrochemical properties and their large chemically active surface. Among other oxide materials, tungsten (VI) oxide (WO_3_) is very promising n-type semiconductor. Due to relevant physical, photoelectrochemical and catalytic properties, WO_3_ is used in lithium-ion batteries [4], electrochromic windows [5], solar energy conversion systems [6,7,8,9,10,11], volatile organic compounds (VOCs) and gas sensors [12]. The sensors based on WO_3_ still suffer from high operating temperature, low sensitivity or high limit of detection (LOD). The improvement of sensitivity and LOD is especially significant for medical purposes, where sensitive and accurate detection is necessary. Therefore, it is very important to find out how different crystallinity and morphology of coatings influence WO_3_ activity. The mechanism of gas sensing is a complex process. It involves adsorption/(catalytic action)/desorption steps, which in air atmosphere are significantly affected by oxygen molecules that are chemisorbed and/or physically-adsorbed on the WO_3_ surface. Depending on the operating temperature, the negatively charged (O^−^, O_2_^−^ and O^2−^) oxygen species on the surface of WO_3_ can be formed. Formation of these species can be presented by following Equations [13,14,15]:O_2(gaseous)_→O_2(adsorbed)_(1)
O_2(adsorbed)_ + e^−^ →O_2_^−^ (<150 °C)(2)
O_2_^−^ + e^−^ →2O^−^(150–400 °C)(3)
O^−^ + e^−^ →O^2−^(>400 °C)(4)

When analyte gas molecules are adsorbed on tungsten oxide, the electrons are either accepted from the surface (oxidizing gas) or donated to the tungsten oxide surface (reductive gas), which leads to an increase or decrease of WO_3_ resistance, respectively. In most cases, the highest catalytic activity of WO_3_-based structures is observed at high temperatures. Working at a high temperature requires more sophisticated and expensive sensor construction and additional equipment for regulation of temperature in order to get a stable sensor response, which eventually leads to high power consumption. Therefore, in recent years, low-temperature gas sensors based on light-activated metal oxide semiconductors have attracted a lot of attention [16,17,18,19,20,21,22,23]. It was suggested that light affects gas sensor performance in the following ways: (1) light influences dissociation of oxygen species and, consequently, the adsorption of VOC-based analyte molecules; (2) light increases density of free electron–hole pairs and facilitates carrier generation, thus enabling the sensor to work at room temperature with high sensitivity and selectivity [16,24]. The gas-sensing properties of metal oxide semiconductors are also known to be strongly influenced by illumination conditions [1,25,26] because light modulates the density of charge carriers. Effective generation, separation and transport of light-generated charge carriers and, consequently, the sensing ability of WO_3_ are highly dependent on the method of synthesis. Various techniques, including hydrothermal [27,28], reactive sputtering [27], spray pyrolysis [29], sol-gel [30,31] and electrodeposition [32,33] are used for the formation of tungsten (VI) oxide layers. Among them, the sol-gel method is very popular because it is easily controllable, does not require any sophisticated procedures or specialized tools and allows adjustment of the most optimal parameters in a simple way [31,34,35,36]. Proper understanding of how some factors influence the properties of semiconductor oxide films is essential for tailoring sensors’ response to light as well as their sensing performance.

The aim of this research was to form WO_3_ layers under different conditions using sol-gel synthesis method and to determine their photo-induced redox properties at room temperature. Crystalline structure, composition, and morphology of WO_3_ coatings were characterized using X-ray diffraction (XRD), scanning electron microscopy (SEM) and Fourier Transform Infrared spectroscopy (FTIR). Photoelectrochemical activity of the coatings was evaluated by cyclic voltammetry, and correlation of the data with the results of time-resolved photoluminescence decay measurements was analysed. 

## 2. Experimental

### 2.1. Chemicals

All chemicals were of ‘Analytical grade’ and were used as received from suppliers, without any further purification. 

Sodium tungstate was purchased from Carl Roth (Karlsruhe, Germany), ammonium oxalate from Chempur (Piekary Śląskie, Poland), hydrochloric acid from Chempur (Piekary Śląskie, Poland), hydrogen peroxide from Chempur (Piekary Śląskie, Poland), ethanol from Reachem (Bratislava, Slovakia), propanol from Reachem (Bratislava, Slovakia), sulfuric acid from Reachem (Bratislava, Slovakia). 

### 2.2. Formation of Photoelectrochemically Active Tungsten Oxide Layers on the FTO and Glass Substrate

WO_3_ thin films on FTO and glass substrate were prepared by sol-gel method in aqueous solution. Firstly, FTO substrates were cut into 1 mm × 2 mm slides and washed under ultrasonication with acetone, ethanol and finally deionized water for 15 min per wash. During the synthesis, a sodium tungstate dihydrate (Na_2_WO_4_ × 2H_2_O) (from Carl Roth) was used as a precursor and ammonium oxalate ((NH_4_)C_2_O_4_; AO) (from Chempur) was used as capping agent. They were dissolved in distilled water and then HCl (from Chempur) was added under continuous stirring for 10 min at 40 °C. Afterwards the hydrogen peroxide (H_2_O_2_) (from Chempur) was added to the above-mentioned solution under continuous stirring for 10 min at 40 °C to obtain peroxotungsten acid (PTA). Further, ethanol (EtOH) (from Reachem) was added as a reductant to the prepared PTA mixture (PTA + EtOH), which was used for the formation of WO_3_ coating. After 10 min, cleaned FTO substrates were dipped in a ‘face-down position’ into the prepared mixture and incubated for 140 min and 180 min. Synthesis was carried out at 85 °C constant temperature in a water bath. After the formation of WO_3_ coating, the slides were rinsed in distilled water for 1 min and then they were dried in the drying oven at 40 °C for 10 h. The same procedures were followed for preparing of PTA and propanol (PrOH) (from Reachem) sol-gels, just instead of ethanol a propanol was added to PTA solution. Finally, samples were annealed at 500 °C for 2 h in ambient atmosphere; heating rate was 1 °C min^−1^ and starting temperature was 20 °C. 

### 2.3. Characterization of Oxide Layers

#### 2.3.1. X-ray Diffraction, Scanning Electron Microscopy

XRD patterns of the films on FTO substrate were obtained using an X-ray diffractometer SmartLab (Rigaku, Oxford, UK) equipped with 9 kW rotating Cu anode X-ray tube. Grazing incidence (GIXRD) method was used in 2θ range at 20–80 °C. An angle between a parallel beam of X-rays and a specimen surface (ω angle) was adjusted to 0.5°. Match software and Crystallography Open Database (COD) was used for phase identification. The average crystallite size D, of each sample was calculated using the Scherrer equation [37,38].

The surface morphology of the tungsten oxide layers on FTO substrate was investigated using Helios NanoLab dual beam workstation equipped with X-Max 20 mm^2^ energy dispersion spectrometer (Oxford Instruments, Oxford, UK). 

The phase composition was investigated by Fourier Transform Infrared spectroscopy using a PerkinElmer spectrophotometer, with a resolution of 4 cm^−1^ over a wavenumber range of 450–4000 cm^−1^. 

#### 2.3.2. Electrochemical Measurements

Voltammetric measurements were performed using three-electrode cell and potentiostat/galvanostat AUTOLAB 302 from Ecochemie (Utrecht, The Netherlands). Tungsten (VI) oxide films deposited on flouride-doped tin oxide (FTO) substrates were used as working electrodes. Silver chloride electrode with saturated KCl solution (Ag/AgCl/sat. KCl) and Pt wire were used as reference and counter electrodes, respectively. The surface of working electrodes was illuminated with high intensity discharge Xe-lamp with 6000 K spectrum and calibrated with a silicon diode to simulate AM 1.5 illumination (∼100 mW cm^−2^) at the sample surface. Experiments were performed in the solution of 0.5 M H_2_SO_4_. Current density values were calculated based on geometric area of the working electrode. 

#### 2.3.3. Time-Resolved Photoluminescence Decay Based Evaluation of WO_3_-Based Coatings

Optical characterization was performed by the evaluation of photoluminescence signal from the samples using a time-correlated single photon counting Edinburgh-F900 spectrophotometer (Edinburg Instruments Ltd., Livingston, UK). The photoluminescence spectra of WO_3_ coatings were excited by solid-state laser with an excitation wavelength of 375 nm (the average pulse power was about 0.15 mW/mm^2^, the pulse duration 76 ps) and measured in the range of 400 to 700 nm. All photoluminescence spectra were corrected for the instrument sensitivity. The photoluminescence decay kinetics was measured with the same Edinburgh-F900 spectrophotometer. The pulse repetition rate was 1 MHz and the time resolution of the setup was about 100 ps taking into account temporal deconvolution procedure.

The position and intensity of the photoluminescence maximum was determined as the corresponding characteristics of Gauss function using the Origin program.

## 3. Results and Discussion

### 3.1. XRD, SEM and FTIR Analysis of WO_3_ Coatings

The crystalline structure and surface morphology of tungsten (VI) oxide coatings were characterized by XRD and SEM techniques. In Figure 1, XRD patterns of samples prepared from PTA + PrOH and PTA + EtOH sol-gels and annealed at 500 °C 2 h are shown. It is obvious that different reductants used in the synthesis and different dipping times influence crystalline structure of the coatings. The main crystalline phase of synthesized coatings in all samples is monoclinic tungsten (VI) oxide. Sample prepared from PTA + PrOH sol-gel with 140 min dipping time has the highest crystallinity, because the peaks investigated in the whole range of 2θ values are the most intensive. Clusters of three peaks at 2θ = 23.20°, 23.60°, 24.29° and two peaks at 2θ = 33.42° and 34.12° are attributed to monoclinic tungsten trioxide (marked with asterisk) in accordance with PDF no. 96-210-6383 of the COD. The sample with 180 min dipping time has less intensive peaks, but a cluster of three peaks, which correspond to (002) (020) (200) facets, is well observed. It is noteworthy that the intensity of facet (200) in this sample decreased dramatically. This means that dipping duration influences the formation of the facets in tungsten trioxide crystallite and lattice parameters as well. Diffractograms of the samples prepared from PTA + EtOH sol-gel exhibit the clusters of three and two peaks. The crystallinity of the coating with 180 min dipping time is slightly higher, because peaks are more intensive.

The facets are the same as in the case of films formed from PTA + PrOH sol-gel. Facet (200) in all the samples has the highest intensity, suggesting selective orientation of the (200) planes parallel to the substrate and [200] direction vertical to the substrate. This can be attributed to the fact that this facet is growing parallel to the FTO substrate [39,40]. Among various crystalline structures of WO_3_, the monoclinic structure is highly stable and the photoelectrochemical activity of the films is known to depend on the facets exposed to solution phase [41]. 

The crystallite size is an important parameter that influences the properties of metal oxide nanostructures; therefore, the crystallites size of all samples was evaluated using Scherrer’s equation: D = kλ/(βcos θ),(5)
where D is the crystallite size, k is a shape factor, λ is the X-ray wavelength (0.15406 nm), β is the full width at half maximum intensity in radians and θ is the Bragg angle. For calculation, the most intensive XRD pattern at 23.20° position was used and results are presented in Table 1. In PrOH-180, EtOH-140 and EtOH-180 coatings 5.58 nm, 5.51 nm and 5.55 nm crystallites were formed, respectively; however, in the coating, PrOH-140 crystallites are approximately 20% bigger and reach 6.57 nm, but in general, the differences in crystallite sizes of the formed samples are insignificant. During the process of synthesis these crystallites combine to form larger particles, i.e., the structural units constituting the coating. Contrary to crystallites, the shape and size of particles was found to differ very significantly depending on synthesis conditions. 

In Figure 2, the SEM images of tungsten trioxide coatings are presented. Morphology of the coatings formed from PTA + PrOH and PTA + EtOH sol-gels differs significantly. Longer synthesis time influences the growth of the WO_3_ H_2_O structure as well as the final crystalline tungsten trioxide morphology after annealing [39]. Coatings synthesized from PTA + PrOH sol-gel are composed of very dense layer of 20–100 nm sized particles with randomly distributed 2 µm size agglomerates. With increasing dipping time (180 min), bigger agglomerates are formed (b). Morphology of the samples prepared from PTA + EtOH sol-gel is completely different (c,d). The coatings are dense, composed of randomly oriented submicrometer-sized (200–1000 nm), vertically aligned plates, which are located very close one to each other forming the network with high surface area. Dipping time does not influence the size of the plates significantly, however the presence of very fine-grained areas can be seen on the surface of “180 min” coating. In both samples, small pores between the particles can be observed. All coatings prepared from PTA + EtOH and PTA + PrOH sol-gels are without cracks in the structure, what facilitates the transfer of photogenerated charge carriers. In accordance with literature data, the difference in the morphology of the coatings can be explained considering the colloidal stability of oxide particles in different solvents as well as reducing ability of different alcohols. It has been reported [42] that reducing ability of primary alcohol increases with decreasing carbon chain length. Thus, ethanol is stronger reductant than propanol and the crystallization of WO_3_
**•**H_2_O should initiate and proceed faster in EtOH-containing medium. On the other hand, the energy barrier, which inhibits the agglomeration of the particles, is directly proportional to the dielectric constant of the liquid medium and the surface potential [43]. In the case of ethanol, the dielectric constant of solution is relatively high and the energy barrier is high enough. Therefore, the primary oxide particles precipitated from this solution are stable, do not agglomerate and their growth proceeds mainly in two directions, resulting in plate-shaped morphology. When propanol containing solvent is used, the crystallization and growth of the WO_3_**•**H_2_O phase is slower and the dielectric constant of solution is lower. Consequently, the primary particles are unstable and prone to agglomeration when attraction forces are predominating against repulsion forces. This leads to nanoparticulate morphology with random distribution of micro- and nano-scale agglomerates. SEM observations are in agreement with XRD results, which are quite similar for “EtOH-140” and “EtOH-180” coatings, but differ significantly between “PrOH-140” and “PrOH-180” as well as between “PTA + EtOH” and “PTA + PrOH” sol-gels. Although crystallite sizes in all the synthesized coatings are almost the same (Table 1), the shapes and sizes of particles differ significantly and play a crucial role in determining the photoluminescence properties and photoelectrochemical performance of WO_3_ films, as shown below.

Fourier transform infrared (FTIR) spectroscopy confirmed the structural composition and the purity of the formed WO_3_. In Figure 3 the broad absorption peaks at <1000 cm^−1^ indicate the presence of pure tungsten oxide. The band at 619 cm^−1^ is attributed to W–O stretching vibration, while bands at 801 and 762 cm^−1^ are attributed to the inter-bridge stretching O-W-O and the corner-sharing mode W-O-W, respectively [44,45]. Tungsten oxide synthesized from PTA + EtOH sol-gel has much sharper peaks due to a more orderly structure with well-expressed W-O stretching vibration. The broad band at 3400 cm^−1^ can be attributed to W-OH stretching vibration, and the peak located at 1625 cm^−1^ corresponds to W-OH bending vibration mode of the adsorbed water molecules [46].

### 3.2. Time-Resolved Photoluminescence Decay Based Evaluation of WO_3_ Coatings

The normalized photoluminescence spectra of four investigated samples measured at room temperature are presented in Figure 4. It is well known that the PL signals of semiconductors are generated by the recombination of photo-induced charge carriers [47]. The photoluminescence signal is characterized by wide non-symmetric maximum in the range of 400–600 nm, cantered at 426 (2.9 eV), 428 (2.9 eV), 436 (2.85 eV) and 445 (2.8 eV) nm for PrOH-140, PrOH-180, EtOH-140 and EtOH-180 samples, respectively. It is clearly seen that PL maximum shifts to the red spectra region in the case of EtOH-140 and EtOH-180 samples, what can be caused by the differences in particle size and shape compared with PrOH-140 and PrOH-180 samples. Such displacement can be attributed to the fact that different shape of particles can lead to an increase in the concentration of surface defects responsible for PL [48]. It is well known that the size and shape of nanomaterial affect their physicochemical properties [49]. Generally, oxygen vacancies are known to be the most common defects. They usually act as radiative centres in the luminescence processes and can serve as deeply trapped holes in the semiconductors [47]; however, other impurities or defects within WO_3_ thin film might also contribute to the emission at 426, 428, 436 and 445 nm [48,50,51,52] observed in this study.

It is necessary to mention that blue photoluminescence band in the 400–440 nm region also could be attributed to optical transition in the isolated -OH groups formed at the WO_3_ coating surface [53,54]. This correlates well with FTIR results, because W-OH stretching and bending vibrations were observed at 3400 cm^−1^ and 1625 cm^−1^ wavenumbers, respectively. This fact is in good agreement with investigations of several metal oxides, which show the same photoluminescence of -OH-based photoluminescence centres, which are present in oxidized nanocrystalline and porous silicon, hydrated alumina oxide, hydrated lead oxide and zinc oxide [55].

The presence of oxygen in the surrounding atmosphere is also important for the generation of photoluminescence and for the catalytic activity of tungsten oxide layer [56,57] as it can interact with oxygen vacancies as represented by the following equations:½ O_2(g)_ + e^−^ + *A*_s_ ↔ O_(ads)_^−^ *A*_s_ is an oxygen adsorption site(6)
O_(ads)_^−^ + *V*_O_^+^ ↔ O_(bulk)_ *V*_O_ is an oxygen vacancy(7)

Oxygen adsorption (Equation (6)) and WO_3_ oxidation (Equation (7)) are happening consecutively and adsorbed/integrated oxygen (Equations (6) and (7)) leads to the formation of electron depletion layer on the surface of WO_3_ film. When adsorbed O_(ads)_^−^ oxygen species are removed from the surface reacting with the reducing gases or organic compounds, then the injection of the electrons is narrowing ‘the width’ of the electron depletion layer, which leads to the decrease in WO_3_ layer resistance. Two different charge transfer mechanisms are observed for WO_3_ layers: the first is based on n-type conductivity that is typical to stoichiometric WO_3_ [58,59,60] and the other mechanism is based on p-type conductivity that is characteristic for non-stoichiometric tungsten oxide WO_3-x_. The latter was reported and can be responsible for the low-temperature sensitivity of WO_3-x_-based layers towards gaseous analytes [61,62].

Time-resolved photoluminescence decay measurements presented in Figure 5 reveal that the average photoluminescence decay time of PrOH-140, PrOH-180, EtOH-140, EtOH-180 samples is 11.2, 11.3, 4.5 and 10.9 ns, respectively (Table 2). It is noteworthy that in the case of sample EtOH-140 the intensity of photoluminescence was the lowest (not shown) and average photoluminescence decay time was the shortest. In addition, it is necessary to mention that, in the case of EtOH-140 sample, no formation of dense nanostructured particles was observed (Figure 2). The reasons for different behaviour of sample EtOH-140 and correlation between PL properties, photoelectrochemical performance and morphological features of WO_3_ films formed under different conditions are discussed in the next section.

### 3.3. Photoelectrochemical Evaluation of WO_3_ Coatings Deposited on FTO-Glass Electrode

Photoelectrochemical behaviour of WO_3_ films deposited on conducting glass/FTO substrate was investigated in the solution of 0.5 M H_2_SO_4_ in dark and under illumination and the results are presented in Figure 6. Cyclic voltammogram (CV) of sample EtOH-140 in the dark (Figure 6a, curve 1) represent a typical response of tungsten (VI) oxide. The increase in cathodic current at E < 0.2 V and the anodic current peak that is observed in the same range of potentials reflects the reversible redox transition between W (VI) and W (V) oxygen species. Under dark conditions, the zone between 0.4 V and 1.8 V is the range of passivity, where no electrochemical processes occur. Under illumination, a remarkable increase of photoanodic current is observed at E > 0.4 V (Figure 6a, curve 2). This is consistent with n-type conductivity of WO_3_ coatings formed by sol-gel technology. In accordance with references [41,63], the main photoanodic process occurring in sulfuric acid solution is the oxidation of HSO_4_^−^ into S_2_O_8_^2^ by holes photogenerated in tungsten (VI) oxide, because the energy of holes is high enough to drive this redox reaction (E^0^(S_2_O_8_^2−^/2HSO_4_^−^) = 2.12 V). The processes of reversible redox transition between W(VI)/W(V) oxygen compounds at E < 0.4 V are not affected by illumination conditions. Comparison of CVs of FTO-supported WO_3_ films formed under different synthesis conditions is presented in Figure 6b. It should be noted here that in the E range below 0.4 V the curves of all samples practically coincide. This means that electrochemically active surface area of differently formed samples is very similar. In terms of photoelectrochemical activity, the sample EtOH-140 significantly outperforms all the rest WO_3_ films, as can be seen from the comparison of cyclic voltammograms at E > 0.4 V. These observations imply that the differences in PEC activity of the samples cannot be attributed to mere difference in their surface area. The photoelectrochemical activity of WO_3_ coatings decreases in the following sequence: EtOH-140 > EtOH-180 > PrOH-140 > PrOH-180. Interestingly, the sample EtOH-140 with the shortest photoluminescence decay time (τ_ave_ = 4.5 ns), i.e., the fastest recombination, exhibits the highest photoelectrochemical activity. Such a discrepancy can be explained by taking into account the morphological features of the samples, as shown in Figure 2. One can see that the sample EtOH-140 is the only one which does not have the very fine-grained nanocrystalline areas discussed above. Most likely, these areas have a very high concentration of grain boundaries, which act as traps for the photogenerated charge carriers. The presence of such traps impedes the radiative recombination of charge carriers, leading to higher values of τ_ave_. From the viewpoint of PEC performance of the samples, these trap states slow the transport of photogenerated charge carriers towards WO_3_/solution interface, resulting in lower photocurrent. Consequently, slower kinetics of recombination due charge carrier trapping correlates with lower photoelectrochemical activity, i.e., the efficiency of charge transfer at the WO_3_/solution interface. The effect of particle size and grain morphology on the movement of photogenerated electrons and holes in semiconductors is well-known [64,65]. In the case of tungsten (VI) oxide, it has been reported [40,66] that films composed of irregular disordered perpendicularly oriented crystallites show higher photoelectrochemical performance than films with nanocrystalline particles. Under conditions of applied external bias (Figure 6), the rate of electron-hole recombination is suppressed, and the morphology of vertically aligned submicrometer sized plates results in the highest photocurrent of sample EtOH-140.

Here, the presented experimental results reveal that using ethanol as a reductant in the sol-gel synthesis applied in this study and 140 min dipping time in PTA + EtOH-based solution ensures the formation of WO_3_ structure, which is optimal for efficient separation and transport of photogenerated charge carriers due to formation of vertically aligned plates. Longer dipping duration or the use of propanol as a reductant favour the formation of a nanocrystalline structure, which is detrimental for fast transport of charge carriers due to presence of numerous grain boundaries which act as traps for photogenerated electrons.

## 4. Conclusions and Future Trends

Layers of tungsten (VI) oxide were formed on conducting glass substrate by sol-gel synthesis method using two different reducing agents and different durations of coating formation. It was found that using propanol as a reductant leads to formation of dense nanostructured WO_3_ films with randomly distributed agglomerates composed of particles of several micrometres size, whereas using ethanol as a reductant favours formation of uniform layers of vertically aligned submicrometer sized plates. Investigations of time-resolved photoluminescence and photoelectrochemical activity of WO_3_ films revealed that the latter microstructure is favourable for the efficient transport of photogenerated charge carriers. Vertically aligned plates were found to quench the PL of WO_3_ coatings as such morphology is characterized by lower intensity and fast decay (4.5 ns) of photoluminescence compared to the films containing fragments of dense nanostructure. These dense nano-structured areas were suggested to be responsible for rather strong photoluminescence in the range of 400–440 nm and photoluminescence decay times of 11 ns. Moreover, numerous grain boundaries present in the nano-structured fragments of WO_3_ coatings contain oxygen vacancies and other defects, which act as traps for the photogenerated charge carriers, slowing down their transport and leading to inferior photo-electrochemical performance. These findings provide guidelines for the synthesis of light-sensitive WO_3_ films, which will be assessed for gas sensing properties at room temperature.

## Figures and Tables

**Figure 1 materials-13-02814-f001:**
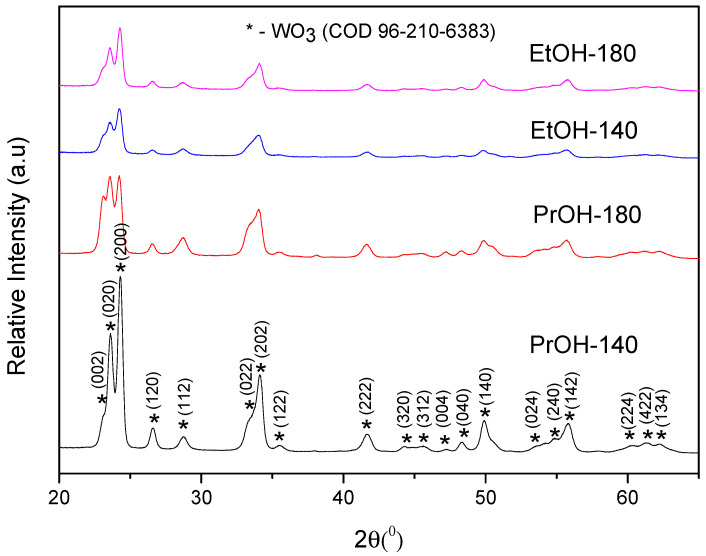
X-ray diffraction (XRD) spectra of WO_3_ layers prepared from PTA + EtOH and PTA + PrOH sol-gels using dipping times of 140 min and 180 min; samples wereannealed at 500 °C for 2 h.

**Figure 2 materials-13-02814-f002:**
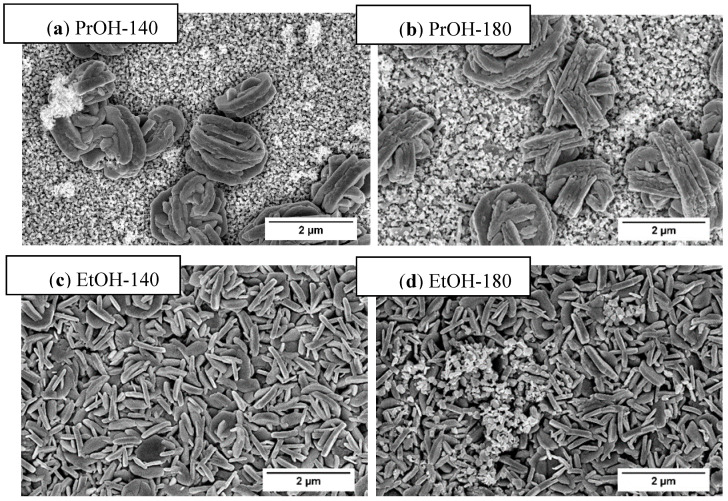
SEM images of WO_3_ sample prepared from PTA + PrOH sol-gel dipped for 140 min (**a**); dipped for 180 min (**b**) and PTA + EtOH sol-gel dipped for 140 min (**c**); dipped for 180 min (**d**) and annealed at 500 °C for 2 h. Magnification ×25,000.

**Figure 3 materials-13-02814-f003:**
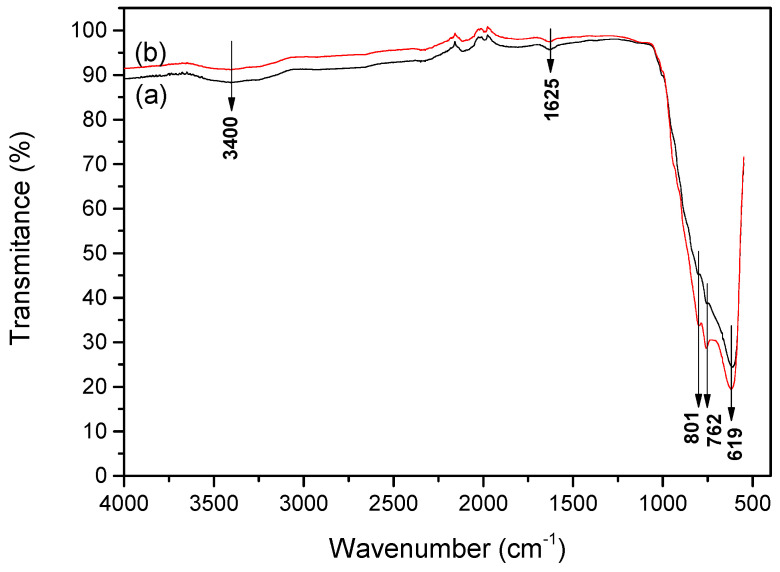
Fourier Transform Infrared (FTIR) spectrum of WO_3_ powder prepared from (**a**) PTA + PrOH sol-gel and (**b**) PTA + EtOH sol-gel and annealed at 500 °C.

**Figure 4 materials-13-02814-f004:**
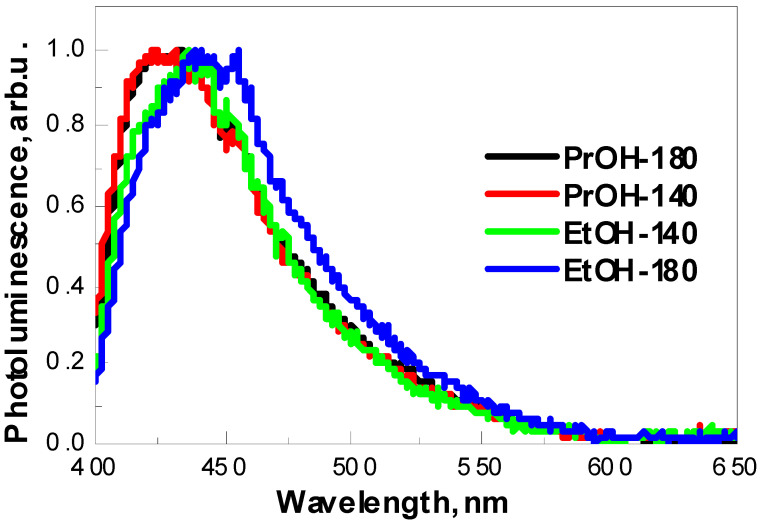
Normalized photoluminescence spectra of samples PrOH-140, PrOH-180, EtOH-140 and EtOH-180 under 375 nm excitation.

**Figure 5 materials-13-02814-f005:**
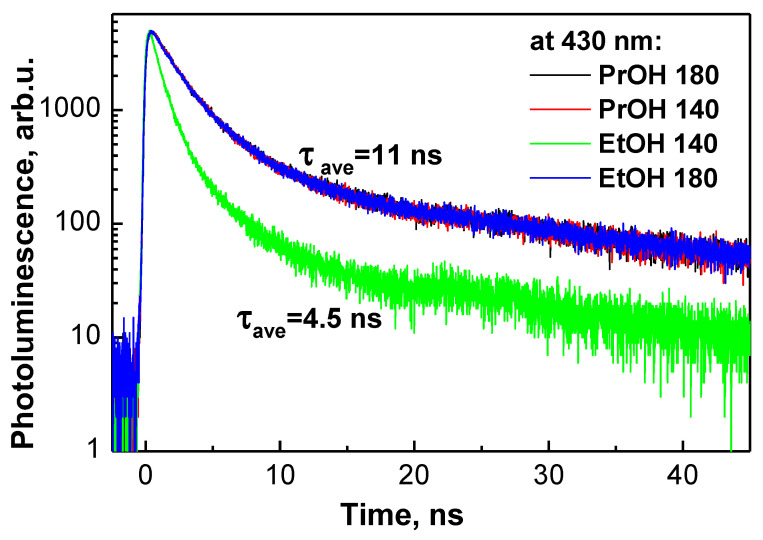
Time resolved photoluminescence decay kinetics of samples PrOH-140, PrOH-180, EtOH-140 and EtOH-180 at 430 nm.

**Figure 6 materials-13-02814-f006:**
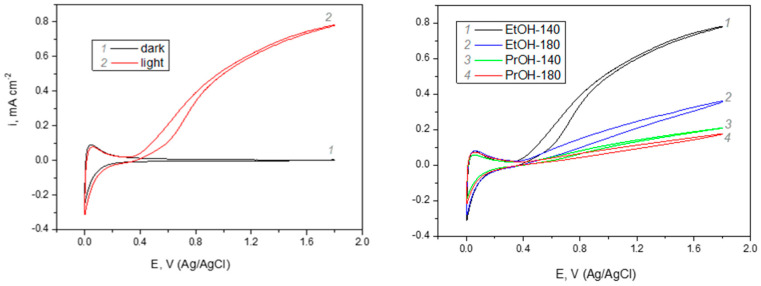
Photoelectrochemical response of FTO/WO_3_ samples formed under different synthesis conditions: (**a**) cyclic voltammograms of sample EtOH-140 in dark and under illumination; (**b**) cyclic voltammograms of indicated samples under illumination; solution 0.5 M H_2_SO_4_, potential scan rate of 50 mV s^−1^, intensity of illumination~100 mW cm^−2^.

**Table 1 materials-13-02814-t001:** Crystallite size of PrOH-140, PrOH-180; EtOH-140 and EtOH-180.

Sample	Coating Formation Time (min)	Crystallite Size (nm)
PrOH-140	140	6.57
PrOH-180	180	5.58
EtOH-140	140	5.51
EtOH-180	180	5.55

**Table 2 materials-13-02814-t002:** Photoluminescence data of WO_3_ samples formed under different conditions.

Sample	λ_em_, nm	τ_1_, ns (%)	τ_2_, ns (%)	τ_3_, ns (%)	τ_ave_, ns
PrOH-140	426	1.6 (37%)	4.1 (32%)	30 (31%)	11.2
PrOH-180	428	1.6 (37%)	4.4 (32%)	30 (31%)	11.3
EtOH-140	436	0.7 (55%)	2.6 (30%)	22 (15%)	4.5
EtOH-180	443	1.5 (33%)	3.9 (36%)	29 (31%)	10.9

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
