# Peer review of "Tuning the Photo-Luminescence Properties of WO3 Layers by the Adjustment of Layer Formation Conditions"

_materials, 2020, doi:10.3390/ma13122814_

Round 1

Reviewer 1 Report

In this article, the authors used a sol-gel synthesis method for the WO3 deposited on the surface of bare glass and glass/FTO. They may use different reductants and the dipping times to control the morphology and crystallite size of the WO3 layers. These WO3 samples were characterized by X-ray diffraction, scanning electron microscopy, Fourier Transform Infrared spectroscopy, photoluminescence, time-resolved photoluminescence and also the photoelectrochemical properties. Although the authors did a lot of characterization works on their prepared WO3 samples, they did not provide sufficient information on the importance of studying in the WO3 layers. Also, they seldomly explain on how the reductants and the dipping times to affect the final structural formation. From their characterizations, it seems that they just send the samples to the instruments without considering the mutual correlations. For example, the crystallite sizes of individual synthesized layers were extracted from the XRD investigation. However, what is the relation or connection to the SEM morphology? The whole article has to be polished to re-submit.

Author Response

Detailed response to Reviewer 1 is in attachment.

Reviewer 2 Report

This work investigates the effect of film formation conditions on the formed crystal structure and resulting photo electrochemical properties looking at varying the reductant and the formation time. It is found that platelet structures grown with ethanol in 140 min outperform both a longer growth time and structures grown with propanol in photo electrochemical testing. The manuscript is well written and logically organised. Conclusions are convincing and explain the presented data. Findings provide valuable information for the development of devices with WO3 films using sol-gel synthesis.

To improve clarity in some areas the authors should consider addressing the following points:

It is mentioned that coatings grown on bare glass slides were also investigated. It is not clear where these were used instead of the FTO slides and a note should be added clarifying which slides are described where that is not the case yet. Any potential differences in films grown on the two substrates, should they exist,  should also be discussed.

It is interesting to see the high degree of similarity between the PrOH and EtOH/180 samples in Figure 5, especially given the differences the other measurements show. Could the authors comment on this?

Could the authors recheck the numberings, at least equation numbering L227f seems to have changed in editing.

Author Response

Detailed response to Reviewer 2 is in attachment.

Reviewer 3 Report

The authors reported the sol-gel synthesis and characterization of WO3 on FTO substrate and their application in PEC. The results are interesting, however, there are some issues need to be addressed before the further consideration.

  1. Why the morphology changed a lot using EtOH in sol-gel process as compared to PrOH?
  2. How the surface area of the samples to affect the photoactivity in PEC?
  3. The authors explained that the shorter PL lifetime of EtOH-140 sample was due to the less surface trap state. How to know the difference of surface trap states in the tested samples? In addition, the TRPL decay of tested samples only can be used to study their kinetics of radiative recombination of charge carriers, which cannot be used to illustrate the charge transfer between WO3/solution interface.

Author Response

Detailed response to Reviewer 3 is in attachment.

Round 2

Reviewer 3 Report

All the issues have been addressed. In my opinion, the current revised version can be accepted to publish on Materials.